# Active Time in Cooperative Activities, Quality of Life and Body Mass Index in Individuals with Intellectual Disabilities. A Model of Structural Equations

Gabriel González-Valero [1], Josep Vidal-Conti [2], Félix Zurita-Ortega [1] and Pere Palou-Sampol [2,*]

1   Department of Didactics of Musical, Plastic and Corporal Expression, University of Granada, 18011 Granada, Spain; ggvalero@ugr.es (G.G.-V.); felixzo@ugr.es (F.Z.-O.)
2   Physical Activity and Sport Science Research, University of the Balearic Islands, 07122 Palma, Spain; josep.vidal@uib.es
*   Correspondence: josep.vidal@uib.cat; Tel.: +34-971172095

**Abstract:** Current research shows that individuals with intellectual disabilities do not engage in enough physical activity to acquire health benefits. However, cooperative learning has been shown to be an effective tool for inclusion and for improving healthy physical habits. The aim of this study is to contrast an explanatory model which incorporates quality of life, active time in cooperative activities, body mass index and age, as well as to analyze, using multi-group structural equations, the existing associations according to the sex of subjects with intellectual disabilities. The convenience sampling used allowed the collection of data from a total of 156 subjects in Granada (Spain), aged between 18–55 years. In terms of gender, the sample was homogeneous, representing 52.6% (*n* = 82) for women and 47.4% (*n* = 74) for men. The active time during the cooperative learning was recorded with the Xiaomi Mi Band 2 activity band, for the quality of life scale (GENCAT) was used, and the body mass index was calculated through its standardized equation. Age was directly associated with body mass index in both sexes. Likewise, age was positively related to the active time of women. Quality of life was directly associated with active time and body mass index was inversely related to active time. This study shows the importance of active time during work and cooperative learning in individuals with intellectual disabilities, as it is associated with an improvement in the quality of life and a reduction in the problems of sedentarism, overweight, and obesity.

**Keywords:** intellectual disability; quality of life; cooperative learning; active time; body mass index

## 1. Introduction

The World Health Organization [1] reports that more than one billion people live with some kind of disability. This represents nearly 15% of the global population. According to data from the Spanish government, of those who have some kind of disability, 8.7% have an intellectual disability (ID) [2]. This group is distinguished by shortcomings in their adaptive behavior and intellectual functioning [3,4]. Those in the group tend to have difficulties acquiring a positive self-image in physical and social respects, and this stems from situations in which they face social isolation, discrimination, and stigmatization [5,6]. In turn, these situations have been associated with more sedentary lifestyles, which can lead to unique health conditions, higher rates of morbidity, and lower life expectancies [7,8].

Against this backdrop, the quality of life of persons with IDs is key to optimally developing their competencies [9,10]. When dealing with the quality of life, various authors reference psychological well-being, social relationships, physical and vital functions, and being able to perform day-to-day activities by oneself [11,12]. This underscores that quality of life is a psychological construct that involves the physical, social, psychological, and functional aspects of well-being that are associated with experiencing life [13,14]. In order to positively influence the factors that affect the quality of life in persons with IDs, resources,

support strategies, interventions, and organizational services must be employed [15,16]. Thus, to improve the quality of life in this group, physical-sports tools involving cooperative work are used to promote good health and active lifestyles [17,18].

It should be emphasized that cooperative learning is one of the most recommended strategies for focusing on inclusion and participation among young disabled persons [19,20]. Performing tasks in which young persons must cooperate and work together in a group to reach objectives is key for securing individual and group responsibilities, properly using social abilities, and creating positive interdependence [21,22]. The methodology behind this kind of learning is one that could be employed in many areas, with physical activity and sport being easy starting points, as they favor cooperative learning [23,24]. It is true that persons with IDs do not meet the minimum recommendations for practicing physical activities proposed by the main international bodies that advocate for health and quality of life [25,26]. However, this is largely related to the high levels of obesity seen in the group; in fact, body mass index (BMI) is considered to be a health indicator for this population [27]. For this reason, many studies have highlighted how important it is for persons with IDs to perform physical activity [28,29], which can lead to both physiological improvements [30] and a better quality of life [31]. In fact, it has been shown that physical activity is an effective tool for preventing social isolation [32,33]. Thus, thanks to recreational physical activity, subjects have been shown to have experienced higher levels of social interaction and participation by simply enjoying the activities and due to the social support received [34,35]. In this way, cooperative physical activity helps to improve and strengthen well-being, quality of life, and the psycho-emotional state by boosting self-esteem, self-confidence, and motivation, which stem from the recreational practice of games and sports [31,36].

In view of the above, we must continue to evaluate the cooperative learning associated with the regular practice of physical-sports activities as a way to promote and improve the quality of life in persons with IDs. Regarding the scientific literature we reviewed, the present research work has the following objectives:

- To test an explanatory model that integrates quality of life, active time spent on cooperative activities, BMI, and age;
- To analyze, using multigroup structural equation modeling, the relationships that exist between quality of life, active time, BMI, and age, based on the sex of subjects with IDs (man/woman).

## 2. Materials and Methods

### 2.1. Design and Participants

The present study employed a non-experimental (ex post facto), descriptive, cross-sectional design to assess persons with IDs. Measurements were taken once from a single group made up of 156 individuals with IDs in the city of Granada, Spain. Participants were aged from 18 to 55 (M = 29.59 ± 9.80), and gender was homogenously represented, with 52.6% of participants being females (N = 82) and 47.4% being males (N = 74). Participant selection criteria included having a recognized level of ID equal to or greater than 33% and not having a motor disability that would make it difficult to practice physical sport activities.

### 2.2. Instruments

The study used (ad hoc) self-reported questionnaires to record sociodemographic data, which included participants' age and sex (either male or female).

To record their active time during cooperative games and activities, we made use of the Xiaomi Mi Band 2, a GPS bracelet that measures health indicators and includes a three-axis accelerometer to determine steps based on the most indicative movement patterns seen in people. In this way, we were able to record the total number of steps and minutes of active time during physical-recreational sessions.



Quality of life was measured with the GENCAT scale [37], which is made up of 69 items related to observable aspects of individual quality of life. Participants responded to these items on a Likert scale with four options ranging from 1 to 4, corresponding to "Never" and "Always," respectively. These items cover eight multidimensional subscales that allowed us to obtain a general measure of the quality of life [38]. The subscales are as follows: emotional well-being (items 1–8), interpersonal relationships (items 9–18), material well-being (items 19–26), personal fulfillment (items 27–34), physical well-being (items 35–42), self-determination (items 43–51), social inclusion (items 52–59), and rights (items 60–69). With regard to the reliability of the scale, the validation study [37] had an $\alpha = 0.920$ and acceptable values for the subscales. In the present study, we found an $\alpha = 0.854$, an acceptable level of reliability.

BMIs were calculated by taking participants' weights in kilograms divided by their squared height in meters. These criteria were recorded using a digital scale (Seca 876) and a stadiometer (Seca 213).

### 2.3. Procedure

A letter and an informed consent form were drafted by the Department of Teaching of Musical, Visual, and Corporal Expression at the University of Granada, and they were sent to various centers so that participants and family members or legal guardians could be informed of the intent of our research. Additionally, monitors from the centers participating in our study informed participants of its purpose. The questionnaires were completed by the subjects with the help of the aforementioned monitors and family members. A total of 36 cases were discarded from the sample because the questionnaires were not properly filled out.

Data were gathered from January to June 2019, during the school day, before classes started. In addition to the researchers being present during the data collection and the cooperative games and activities, family members and monitors from the centers were given instructions to help guide them and resolve any possible doubts.

Our study is in line with the ethical principles for research established by the Declaration of Helsinki [39], ensuring anonymity and respecting participants' rights. Additionally, the Ethics Committee of the University of Granada approved this research (1230/CEIH/2020).

### 2.4. Data Analysis

To analyze the data obtained, we used SPSS 24.00 statistical software (IBM Corp., Armonk, NY, USA) in order to establish the frequencies and averages of the basic descriptive analysis. We used Cronbach's coefficient to determine the internal consistency of the instruments, finding a reliability index of 95%. The multigroup structural equation modeling (SEM) was carried out using AMOS 23.0 software (IBM Corp., Armonk, NY, USA). SEM was used to establish the relationships between the variables included in our theoretical model (Figure 1) for both groups (men and women). From here, we built a general model for the study population along with two differentiated models in order to confirm the relationships between the variables as a function of participants' sex. The SEM in our analysis was performed with four observable variables that provided explanations for the relationships. In this case, the causal explanations of the endogenous variables were made considering the relationships observed between the indicators and the reliability of the measurements. In this way, errors made in measuring the observable variables were included in the model and could be directly controlled for and interpreted as coefficients in the multivariate regression. The one-way arrows represent the influence lines between variables, and they are interpreted based on their regression weights. Furthermore, we established a level of significance of 0.05 using Pearson's chi-squared test.

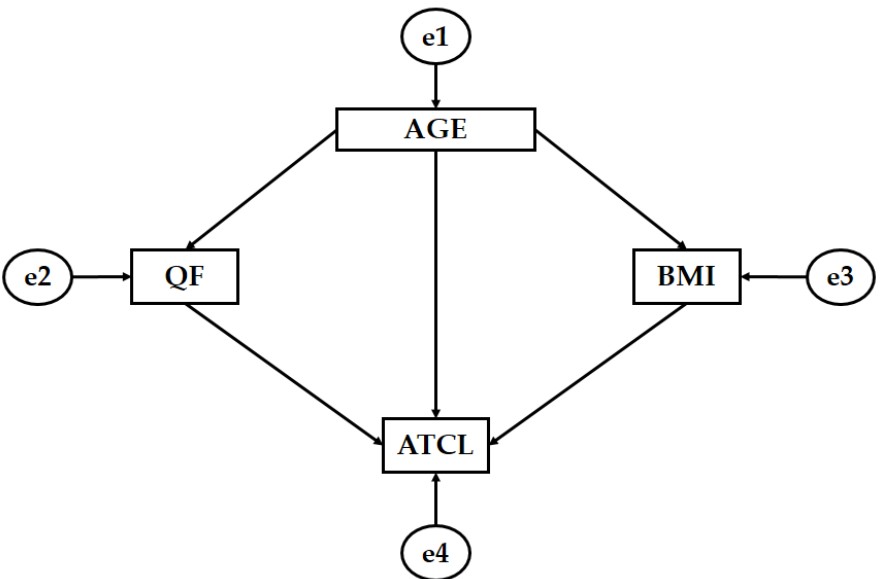

**Figure 1.** The Theoretical Model. Note: Quality of Life (QL); Body Mass Index (BMI); Active Time in Cooperative Learning (ATCL); Age (AGE).

Active time during cooperative learning (ATCL) is an endogenous variable that is affected by the quality of life (QF), age, and BMI. The QF and BMI also act as endogenous variables that are affected by age.

We analyzed the fit of the model to verify compatibility between the model created and the empirical data obtained. We assessed the reliability of the model based on goodness of fit, focusing on the criteria proposed by Marsh (2007) [40]. For the chi-squared test, values associated with a *p*-value that is not significant indicate an acceptable goodness of fit, however, we must also consider other similar indices, as this particular statistic is very sensitive to the effects of sample size [41]. So, we used other parameters such as the comparative fit index (CFI), the normal fit index (NFI), the incremental fit index (IFI), and the Tucker-Lewis fit index (TLI). To have an acceptable fit, values must be above 0.90, with values over 0.95 representing an excellent fit. We also used the root mean square error approximation (RMSEA), for which values below 0.08 are considered to represent an acceptable fit and values below 0.05 are considered to represent an excellent fit.

### 3. Results

The model we developed via a consideration of the variables measured in persons with IDs as a function of sex shows a good fit in all assessment indices. The chi-squared test returns a significant *p*-value ($X2 = 64,382$; df: 17; $p < 0.001$). However, this index cannot be understood straightforwardly, as it is susceptible to and influenced by sample size [40]. To address this issue, we used other indices of the goodness of fit that are less sensitive to sample size.

The CFI analysis returned a value of 0.998, which suggests an excellent fit. The NFI analysis returned a value of 0.973, the IFI was 0.998, and the TLI was 0.987, all suggesting an excellent fit. The RMSEA also indicated an excellent fit, with a value of 0.022.

The regression weights for the general theoretical model can be seen in both Figure 2 and Table 1, and relationships are seen to be statistically significant at the $p < 0.05$, $p < 0.01$, and $p < 0.001$ levels. Age is positively correlated with BMI ($p < 0.001$; r = 0.445)—with a regression weight of medium power—and with ATCL ($p < 0.05$; r = 0.275)—with low-medium power. Similarly, quality of life is positively correlated with ATCL ($p < 0.01$; r = 0.284) with a low-medium power. Finally, BMI is negatively correlated with ATCL ($p < 0.001$; r = −0.411), and its regression weight has medium power.

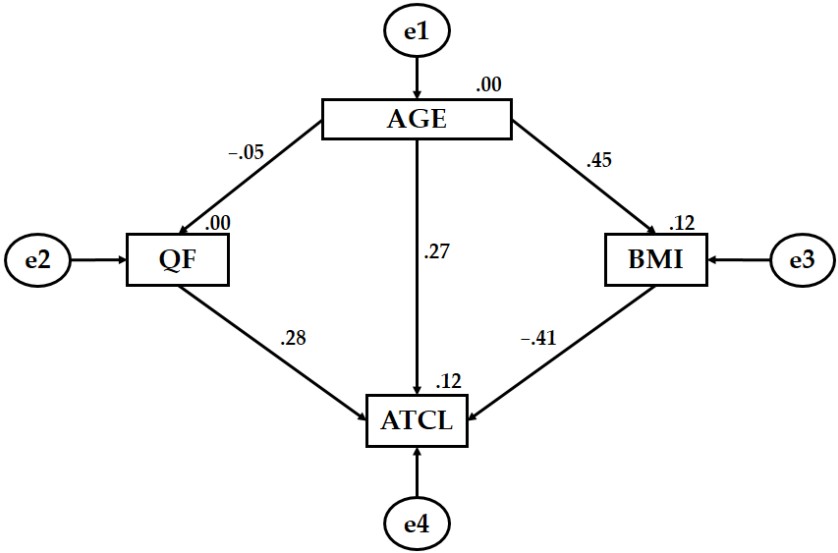

**Figure 2.** The structural equation for the theoretical model. Note: Quality of Life (QL); Body Mass Index (BMI); Active Time in Cooperative Learning (ATCL); Age (AGE).

**Table 1.** The structural model for the theoretical model.

| Associations between Variables | | | R.W. | | | | S.R.W. |
|---|---|---|---|---|---|---|---|
| | | | Estimations | S.E. | C.R. | P | Estimations |
| QL | ← | AGE | −0.001 | 0.002 | −0.578 | 0.563 | −0.046 |
| BMI | ← | AGE | 0.163 | 0.036 | 4.580 | *** | 0.445 |
| ATCL | ← | AGE | 0.132 | 0.061 | 2.174 | * | 0.275 |
| ATCL | ← | QL | 7.229 | 2.960 | 2.442 | ** | 0.284 |
| ATCL | ← | BMI | −0.499 | 0.129 | −3.873 | *** | −0.411 |

Note 1: Regression Weights (R.W.); Standardised Regression Weights (S.R.W.); Estimation Error (S.E.); Critical Ratio (C.R.). Note 2: Quality of Life (QL); Body Mass Index (BMI); Active Time in Cooperative Learning (ATCL); Association between variables (←). Note 3: $p < 0.05$ (*); $p < 0.01$ (**); $p < 0.001$ (***).

In Figure 3 and in Table 2 we show the regression weights for the parameters of the theoretical model for men with IDs. In analyzing the indicators of each one of the variables in men, only age was seen to be positively correlated with BMI ($p < 0.001$; $r = 0.427$)—with a regression weight of medium power. No statistically significant relationships were found among the rest of the variables ($p > 0.05$).

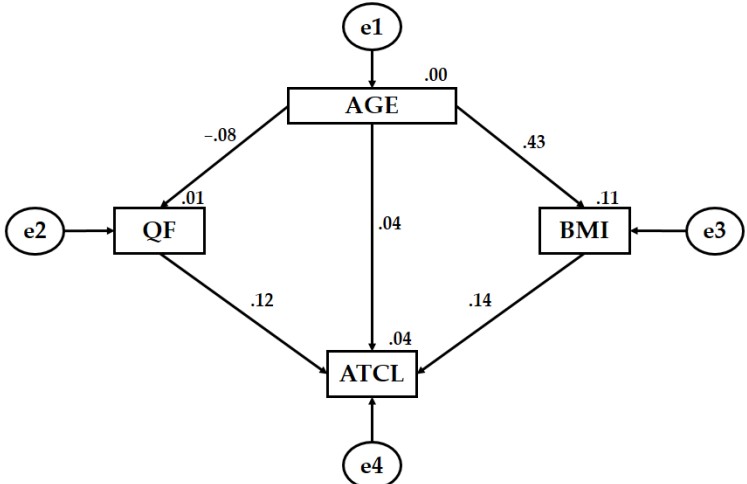

**Figure 3.** The structural equation for the males. Note: Quality of Life (QL); Body Mass Index (BMI); Active Time in Cooperative Learning (ATCL); Age (AGE).

**Table 2.** The structural model for the males.

| Associations between Variables | | | R.W. | | | | S.R.W. |
|---|---|---|---|---|---|---|---|
| | | | Estimations | S.E. | C.R. | P | Estimations |
| QL | ← | AGE | −0.002 | 0.003 | −0.667 | 0.505 | −0.078 |
| BMI | ← | AGE | 0.138 | 0.047 | 2.953 | *** | 0.427 |
| ATCL | ← | AGE | 0.028 | 0.091 | 0.311 | 0.756 | 0.038 |
| ATCL | ← | QL | 4.071 | 3.955 | 1.029 | 0.303 | 0.119 |
| ATCL | ← | BMI | 0.249 | 0.214 | 1.167 | 0.243 | 0.142 |

Note 1: Regression Weights (R.W.); Standardised Regression Weights (S.R.W.); Estimation Error (S.E.); Critical Ratio (C.R.). Note 2: Quality of Life (QL); Body Mass Index (BMI); Active Time in Cooperative Learning (ATCL); Association between variables (←). Note 3: $p < 0.001$ (***).

Both Figure 4 and Table 3 show the regression weights for the parameters of the theoretical model for women, in which statistically significant relationships were found at the $p < 0.001$ level. Age is positively correlated with BMI ($p < 0.001$; r = 0.428)—showing regression weights of medium power. Age is positively and directly correlated with ATCL ($p < 0.001$; r = 0.389), having a low-medium power. Along this line, QF is seen to be positively correlated with ATCL ($p < 0.001$; r = 0.412), with a medium-power regression weight. BMI, however, is negatively and indirectly correlated with ATCL ($p < 0.001$; r = −0.682), with a high-power regression weight

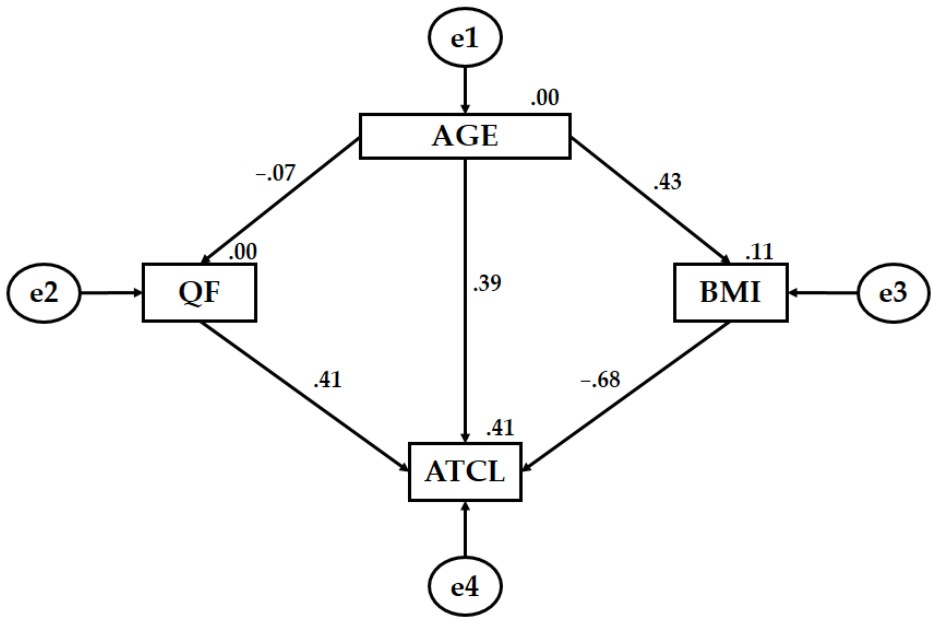

**Figure 4.** The structural equation for the females. Note: Quality of Life (QL); Body Mass Index (BMI); Active Time in Cooperative Learning (ATCL); Age (AGE).

**Table 3.** The structural model for the females.

| Associations between Variables | | | R.W. | | | | S.R.W. |
|---|---|---|---|---|---|---|---|
| | | | Estimations | S.E. | C.R. | P | Estimations |
| QL | ← | AGE | −0.001 | 0.002 | −0.595 | 0.563 | −0.066 |
| BMI | ← | AGE | 0.164 | 0.053 | 3.120 | *** | 0.428 |
| ATCL | ← | AGE | 0.227 | 0.071 | 3.185 | *** | 0.389 |
| ATCL | ← | QL | 14.322 | 3.936 | 3.639 | *** | 0.412 |
| ATCL | ← | BMI | −0.912 | 0.142 | −6.419 | *** | −0.682 |

Note 1: Regression Weights (R.W.); Standardised Regression Weights (S.R.W.); Estimation Error (S.E.); Critical Ratio (C.R.). Note 2: Quality of Life (QL); Body Mass Index (BMI); Active Time in Cooperative Learning (ATCL); Association between variables (←). Note 3: $p < 0.001$ (***).

## 4. Discussion

The present study was carried out on persons with intellectual disabilities because of the emerging concern about the cardiovascular and physical-inactivity problems suffered by this population. The objective of the research was to develop a structural equation model to explain the relationship between the age, BMI, active time, and quality of life of these subjects. To do this, we created a multigroup system in order to compare the relationships between these variables by sex in persons with this kind of disability so that we could paint a more accurate picture of the actual situation. Similarly, we highlight some other research that has had a similar focus [42–47].

By and large, it has been found that as they age, subjects' BMIs increase, and this measure is accepted as a tool for diagnosing issues related to obesity, the potential progression of diseases, and health in general [48]. This finding could be due to the changes that take place during adulthood, including changing eating habits—namely an increased intake of saturated fats—as well as the availability of technological resources that lead to a reduction in the practice of physical activities and thus an increase in BMI [49].

Furthermore, with regard to the relationship between age and time spent exercising, we found that as subjects with IDs age, they dedicate more time to participating in cooperative physical activities. This could be the result of the well-being that physical activity provides, the benefits that exercise can lead to, a greater awareness of the importance of exercise for one's health, or because it is an important source of social interaction [50,51].

Along this line, it is important to note that the time investment in active social activities benefits these subjects' quality of life. This finding is similar to a study [45] that found that performing physical activities contributed to eliminating anxiety, improving physical fitness, establishing social relationships, and promoting personal autonomy in persons with IDs [52]. Additionally, it improves subjective well-being and functional abilities, and it boosts cardiovascular endurance, strength, and muscle balance, all of which means an increase in the quality of life of these subjects [53].

However, we found that subjects who spend less time performing physical activities have greater BMIs, which could affect their health and their quality of life. This finding could be related to the practice and enjoyment of more sedentary activities and to the barriers that persons with IDs face when it comes to practicing physical activities, which could lead to their BMIs increasing with age in a negative way [54]. In fact, the practice of physical activity leads to an improvement of muscle endurance and strength, these factors being crucial in those subjects with a sedentary life and higher BMI indices [55].

With regard to differences by sex, we found that both men and women saw differences with respect to age and BMI, namely, that as these subjects age, they see better BMI values. These data are in line with the data obtained in a study [56] in which they found the lowest BMI values in subjects aged 60 and older. They also found that this cohort was the one that most participated in physical activities. Along this line, authors [57] have highlighted the need to implement programs that promote cooperative physical activity in populations with IDs, beginning in early adulthood so as to prevent problems with obesity, diabetes, and hypertension, among others.

With increases in age, we saw that women spend greater amounts of time participating in cooperative physical activities. This finding is supported by research [58] that found that women with IDs are more active than their male counterparts, and they spend more time being active as the years go by thanks to the well-being provided. However, some studies [59] suggest that women with this kind of disability demonstrate a greater enjoyment of sedentary activities, which could lead to negative effects on the health of this population.

Similarly, females have shown that making use of their time to participate in social, cooperative physical exercise facilitates and improves their quality of life. So, practicing physical activities positively contributes to persons with IDs gaining greater independence and making their own choices about which day-to-day tasks to complete, and it promotes social involvement [60]. That is to say, physical activity leads to improved interpersonal

relationships and social inclusion, and thus produces subjective well-being at an emotional, physical, and material level, all of which have a positive effect on the quality of life.

With regard to the relationship between BMI and active time, we found that when women spend less time on active exercise, their BMI parameters worsen, sedentary behaviors increase, and with this, their quality of life decreases [61]. Additionally, inactivity in women with IDs could promote the appearance of coronary diseases and obesity. To this end, some authors [56,62] have found that women with IDs who do not do any kind of physical activity have shorter life expectancies.

On another note, our study is not without its shortcomings, as it is a descriptive, cross-sectional study, which only enables us to interpret the study variables at the moment at which they were measured. And, the variables were only measured in a small number of subjects, which means that our results cannot be applied to the entire population with IDs. For practical applications, the results we obtained are important, as they show that active time during cooperative learning via physical-leisure activities improves the quality of life and reduces the BMI in individuals with IDs. Thus, we recommend that physical-sports intervention studies based on cooperative learning be carried out, given the benefits that they can produce and so they may be used by the education and science communities. Along this line, it would be interesting to include other variables that could condition participation in physical activities and the quality of life of these subjects. Such variables might include motivation, adhering to a Mediterranean diet, or social relationships.

## 5. Conclusions

The goodness of fit indices for the model we developed for persons with IDs was excellent, which underscores the veracity and importance of the results obtained in our study. Age was a factor that was directly related to the BMI values obtained. Of special note is that age could be a risk indicator for cardiovascular problems and sedentary lifestyles in both sexes. However, age was seen to be positively related to active time during cooperative activities, possibly even more so in women.

Accordingly—both in our general theoretical model and specifically for women with IDs—the quality of life was seen to be directly related to active time during cooperative learning. This finding underscores the importance of these activities and suggests that involving subjects in such activities improves their quality of life. Along this line, we found a strong inverse relationship between BMI and active time, which proves that greater active participation in physical-leisure activities reduces weight problems and obesity.

The present study highlights the importance of active time during cooperative work and learning for persons with IDs, as it is associated with improving their quality of life and reducing physical inactivity, weight problems, and obesity.

**Author Contributions:** Conceptualization, G.G.-V., J.V.-C., F.Z.-O. and P.P.-S.; methodology, G.G.-V. and F.Z.-O.; software, G.G.-V. and F.Z.-O.; formal analysis, G.G.-V. and F.Z.-O.; investigation, G.G.-V., J.V.-C., F.Z.-O. and P.P.-S.; resources, G.G.-V., J.V.-C., F.Z.-O. and P.P.-S.; data curation, G.G.-V. and F.Z.-O.; writing—original draft preparation, G.G.-V. and F.Z.-O.; writing—review and editing, G.G.-V., J.V.-C., F.Z.-O. and P.P.-S.; visualization, G.G.-V., J.V.-C., F.Z.-O. and P.P.-S.; supervision, G.G.-V., J.V.-C., F.Z.-O. and P.P.-S. All authors have read and agreed to the published version of the manuscript.

**Funding:** This research received no external funding.

**Data Availability Statement:** The data presented in this study are available on request from the corresponding author.

**Conflicts of Interest:** The authors declare no conflict of interest.

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
