# Peer review of "Active Time in Cooperative Activities, Quality of Life and Body Mass Index in Individuals with Intellectual Disabilities. A Model of Structural Equations"

_sustainability, doi:10.3390/su13042341_

Round 1

Reviewer 1 Report

Line 310-311 - The reader cannot understand the statement about a section that is not mandatory while this statement is put in the middle of the section and does not seem related to previous or following text. I recommend either to better explain the sentence or delete it completely.

Author Response

We appreciate the time taken to review this manuscript and the comments made, which we believe to be critical for producing rigorous and quality research. Below are detailed all the changes made in the original paper: sustainability-1120558

Amendments have been made in the original manuscript following reviewer's comments. Every amendment contains both the original comment, which page and line numbers are added, and change made in response to that comment. Changes made are colored to be easily identified.

MODIFICATIONS

REVIEWER 1

Comment 1.

Line 310-311 - The reader cannot understand the statement about a section that is not mandatory while this statement is put in the middle of the section and does not seem related to previous or following text. I recommend either to better explain the sentence or delete it completely.

Response 1.

Thank you very much for your comment and for the time spent in reviewing the manuscript. Based on your comment, the statement made in lines 310-311 has been removed.

Reviewer 2 Report

The introduction provides sufficient background to justify the relevance and the need for the study; the rationale is clear.

The present data requires attention. The number of subjects presented in the Abstract part (see lines 17-19)  does not correspond with the numbers presented in the Design and participants part (see lines 86-88).

It is already known that less physical activity directly impacts BMI increase, so how essential it is in the present research the finding presented in lines 264-265.

Some of the claims (see lines 312-323) are generally expected. Thus the Discussion part requires more in-depth, and the epistemological strength of the statements is mitigated somehow.

Author Response

We appreciate the time taken to review this manuscript and the comments made, which we believe to be critical for producing rigorous and quality research. Below are detailed all the changes made in the original paper: sustainability-1120558

Amendments have been made in the original manuscript following reviewer's comments. Every amendment contains both the original comment, which page and line numbers are added, and change made in response to that comment. Changes made are colored to be easily identified.

MODIFICATIONS

REVIEWER 2

Comment 1.

The introduction provides sufficient background to justify the relevance and the need for the study; the rationale is clear.

Response 1.

Thank you very much for your comments and for appreciating the work done in this section.

Comment 2.

The present data requires attention. The number of subjects presented in the Abstract part (see lines 17-19) does not correspond with the numbers presented in the Design and participants part (see lines 86-88).

Response 2.

Thank you very much for your comment. We understand that there may have been some errors during the English style revision. The data in the abstract have been modified and adapted to the method.

Comment 3.

It is already known that less physical activity directly impacts BMI increase, so how essential it is in the present research the finding presented in lines 264-265.

Response 3.

Thank you very much for your comments. Apart from emphasising it as you say, it is justified in the discussion and appears in the conclusions. Thank you again for highlighting these aspects of the manuscript.

Comment 4.

Some of the claims (see lines 312-323) are generally expected. Thus the Discussion part requires more in-depth, and the epistemological strength of the statements is mitigated somehow.

Response 4.

Thank you very much for your comments. A distinction has been made between the discussion and conclusion sections. Thanks to your comments we have realised that the differentiation between the two sections was not clear. In fact, you report that the conclusions section should be more justified. In this case, both sections have been correctly marked and we consider that in the conclusions statements have been made on the basis of the results and justifications of the discussion.

In this case, we consider the justification to be appropriate, taking into account the study population covered by this research.

Thank you for your time and effort in revising the manuscript, which has improved its format and scientific soundness.

Reviewer 3 Report

It should be noted that the introduction provides sufficient background and includes all relevant references. The research design is very appropriate and the methods are adequately described. The results are presented clearly and
extensively, which strongly supports the conclusions of the work.

Author Response

We appreciate the time taken to review this manuscript and the comments made, which we believe to be critical for producing rigorous and quality research. Below are detailed all the changes made in the original paper: sustainability-1120558

Amendments have been made in the original manuscript following reviewer's comments. Every amendment contains both the original comment, which page and line numbers are added, and change made in response to that comment. Changes made are colored to be easily identified.

MODIFICATIONS

REVIEWER 3

Comment 1

It should be noted that the introduction provides sufficient background and includes all relevant references. The research design is very appropriate and the methods are adequately described. The results are presented clearly and
extensively, which strongly supports the conclusions of the work.          

Response 1

Thank you very much for your comments and for your time in reviewing this manuscript.